# PhosphoLipidome Alteration Induced by *Clostridioides difficile* Toxin B in Enteric Glial Cells

**DOI:** 10.3390/cells13131103

**Published:** 2024-06-26

**Authors:** Sandra Buratta, Lorena Urbanelli, Roberto Maria Pellegrino, Husam B. R. Alabed, Raffaella Latella, Giada Cerrotti, Carla Emiliani, Gabrio Bassotti, Andrea Spaterna, Pierfrancesco Marconi, Katia Fettucciari

**Affiliations:** 1Department of Chemistry, Biology and Biotechnology, University of Perugia, Via del Giochetto, 06123 Perugia, Italy; sandra.buratta@unipg.it (S.B.); lorena.urbanelli@unipg.it (L.U.); roberto.pellegrino@unipg.it (R.M.P.); husambr.alabed@dottorandi.unipg.it (H.B.R.A.); raffaella.latella@dottorandi.unipg.it (R.L.); giada.cerrotti@dottorandi.unipg.it (G.C.); carla.emiliani@unipg.it (C.E.); 2Centro di Eccellenza sui Materiali Innovativi Nanostrutturati (CEMIN), University of Perugia, Via del Giochetto, 06123 Perugia, Italy; 3Department of Medicine and Surgery, Gastroenterology, Hepatology & Digestive Endoscopy Section, University of Perugia, Piazzale Lucio Severi 1, 06132 Perugia, Italy; gabassot@gmail.com; 4Santa Maria Della Misericordia Hospital, Gastroenterology & Hepatology Unit, Piazzale Menghini 1, 06129 Perugia, Italy; 5School of Biosciences and Veterinary Medicine, University of Camerino, 62024 Macerata, Italy; 6Department of Medicine and Surgery, Biosciences & Medical Embryology Section, University of Perugia, Piazzale Lucio Severi 1, 06132 Perugia, Italy; pierfrancesco.marconi@outlook.it

**Keywords:** *Clostridioides difficile* Toxin B, enteric glial cells, apoptosis, necrosis, lipidomic analysis, lipid profiles, lipid metabolism pathways

## Abstract

*Clostridioides difficile* (*C. difficile*) is responsible for a spectrum of nosocomial/antibiotic-associated gastrointestinal diseases that are increasing in global incidence and mortality rates. The *C. difficile* pathogenesis is due to toxin A and B (TcdA/TcdB), both causing cytopathic and cytotoxic effects and inflammation. Recently, we demonstrated that TcdB induces cytopathic and cytotoxic (apoptosis and necrosis) effects in enteric glial cells (EGCs) in a dose/time-dependent manner and described the underlying signaling. Despite the role played by lipids in host processes activated by pathogens, to counter infection and/or induce cell death, to date no studies have investigated lipid changes induced by TcdB/TcdA. Here, we evaluated the modification of lipid composition in our in vitro model of TcdB infection. Apoptosis, cell cycle, cell viability, and lipidomic profiles were evaluated in EGCs treated for 24 h with two concentrations of TcdB (0.1 ng/mL; 10 ng/mL). In EGCs treated with the highest concentration of TcdB, not only an increased content of total lipids was observed, but also lipidome changes, allowing the separation of TcdB-treated cells and controls into different clusters. The statistical analyses also allowed us to ascertain which lipid classes and lipid molecular species determine the clusterization. Changes in lipid species containing inositol as polar head and plasmalogen phosphatidylethanolamine emerged as key indicators of altered lipid metabolism in TcdB-treated EGCs. These results not only provide a picture of the phospholipid profile changes but also give information regarding the lipid metabolism pathways altered by TcdB, and this might represent an important step for developing strategies against *C. difficile* infection.

## 1. Introduction

*Clostridioides difficile* (formerly *Clostridium difficile*, *C. difficile*) [1,2] is an anaerobic, spore-forming bacterium, gram-positive, responsible for 15–25% of all opportunistic gastrointestinal infections and >90% of mortality following these infections [3,4,5]. The clinical manifestations of *C. difficile* infection (CDI) range from diarrhea to pseudomembranous colitis, toxic megacolon, colonic perforation, and death [6,7].

CDIs have become an increasingly important health problem because of the following factors: (a) the appearance of new hypervirulent *C. difficile* strains [8,9,10], (b) the increase in the spread of *C. difficile* in the environment [10,11,12], and (c) the improvement in the ability to colonize humans [6,7,10,11,12]. In addition, many patients display recurrent CDI, with up to 35% of them relapsing and up to 60% of them experiencing multiple relapses [13,14].

*C. difficile* produces three different toxins, toxin A (TcdA), toxin B (TcdB), and a binary toxin called *C. difficile* transferase [15,16,17]. The major pathogenic contribution is due to TcdA and TcdB [15,16,17]. However, TcdB is approximately 1000 times more potent than TcdA in terms of cytotoxic activity on cultured cells and is responsible for the major effects of CDI [18,19,20]. Upon cell membrane receptors binding and internalization, TcdA and TcdB induce in all cell types the glucosylation and inactivation of Rho-GTPase with the following major effects: (a) cytopathic effects (early cytoskeleton disruption, cell rounding, cell-cycle arrest); (b) cytotoxic effects, by apoptosis, necrosis or pyknosis, depending on the type of Tcd receptor used and Tcd concentration; and (c) production/secretion of chemokines/proinflammatory cytokines [15,16,17,21,22,23,24]. However, TcdA and TcdB may also cause necrosis and pyknosis in a glucosylation-independent manner [15,16,17,20,22,23,24,25].

Efforts to develop therapies outside the use of antibiotics, such as antibodies [26,27], vaccines [28,29], polymeric binders [30], inhibitors of spore germination [31,32], non-toxigenic *C. difficile* [33], fecal transplantation [34,35], *C. difficile* bacteriophages [36,37], and modified inositol hexakisphosphate (IP6) inactivating TcdB before its uptake [38], have showed limited effects or still present technical limitations [39,40,41,42]. Therefore, further study of TcdB pathogenicity mechanisms is necessary to find key targets to inhibit.

Most in vitro studies with Tcds have focused on the induction of the cytopathic and cytotoxic (apoptosis and necrosis) effects by TcdA and TcdB and on the underlying mechanisms [15,16,17,21,22,23,24]. The studies of apoptosis induction have highlighted the central role of various families of proteases, such as caspases, calpains, and cathepsins, with a role of the mitochondrion [15,16,17,21,22,23,24,43]. Fettucciari et al. demonstrated that TcdB induces cytopathic and cytotoxic effects in Enteric glial cells (EGCs) [21,43,44]. Apoptosis induced by TcdB in EGCs is mediated by the simultaneous activation of three apoptotic pathways [21,43], two involving calpains and caspases and occurring in both a caspase-dependent manner and a caspase-independent manner [21,43], and the third activated by cathepsin B and proceeding in a completely caspase-independent manner [43].

Proteomic studies investigating the molecular changes caused by Tcds demonstrated that in epithelial cell lines CaCo-2 cells and Hep-2, TcdA and TcdB affected the expression of proteins linked to cytoskeleton organization, cell–cell contacts, adhesion, endocytosis, translation/metabolic processes, and intracellular signaling cascades, as well as proteins of specific cellular compartments (e.g., mitochondria) [45,46,47,48,49]. Moreover, using a TcdB mutant lacking the glucosyltransferase domain (TcdBNXN) for Hep-2 cell proteome analysis, it was demonstrated that the glucosyltransferase domain is central only for changes in “cell-cycle” and “chromatin”-associated proteins [49]. Recently, a phosphoproteomic study on the impact of TcdB, TcdB mutant (TcdBNXN), and TcdA on Hep-2 phosphoproteome demonstrated that RAS plays a central role as upstream regulator of the glucosyltransferase-independent effect of TcdB and apolipoprotein C-III on TcdB-induced inflammatory effects, thus elucidating the underlying mechanism of necrosis (pyknosis), a glucosyltransferase-independent effect of TcdB [50].

Until now, a study of lipidome to understand the effects of Tcds on target cell lipid-mediated pathways and structural lipids has never been performed, despite the key role played by lipids in the organization and dynamics of plasma membrane microdomains, membrane trafficking, cellular signaling, regulating cell survival and death, and apoptosis. These cell lipid properties make them an appealing target for pathogens/toxins, in order to modulate host cell processes to their advantage [51,52,53,54,55].

On this basis, we have performed a lipidomic analysis to evaluate changes in the lipid asset occurring during TcdB infection of EGCs, key cells for the physiopathology of the colon [56,57], and one of the main sites of CDI [6,7]. To this purpose, we evaluated the modification of lipid composition in EGCs infected with two concentrations of TcdB, which we have previously demonstrated to induce dose- and time-dependent cytopathic and cytotoxic (apoptosis) effects [21,43,44]. Lipid analysis was carried out by liquid chromatography/mass spectrometry (LC/MS) and a lipid dataset processed using bioinformatics tools, i.e., MetaboAnalist [58] and LipidOne [59]. 

Results of this study demonstrate that TcdB induces in EGCs strong and relevant modification of phospholipid profiles, in terms of lipid classes and molecular species, and suggests lipid pathway alterations possibly responsible for the observed lipidome changes. 

## 2. Materials and Methods

### 2.1. TcdB

TcdB was isolated from *C. difficile* strain VPI10463 and acquired from Enzo Life Sciences (BML-G150-0050; Farmingdale, NY, USA). TcdB was reconstituted to 200 µg/mL and kept at −80 °C as indicated by the manufacturer. Prior to use, it was diluted at 0.1 ng/mL and 10 ng/mL concentration, then immediately used [21,43,44].

### 2.2. EGC Culture and Intoxication with TcdB

The rat-transformed enteroglial cell line (EGC) (EGC/PK060399egfr; ATCC CRL-2690) was purchased from ATCC, Manassas, VA, USA [60] and were cultured in a 75 cm^2^ tissue culture flask, for no more than 20 passages, in Dulbecco’s modified Eagle’s medium (DMEM) with L-glutamine (Sigma Aldrich, Milano, Italy) supplemented with 10% heat-inactivated FBS (Invitrogen, Milano, Italy) and 1× penicillin/streptomycin (Invitrogen) (complete growth medium) at 37 °C with 5% CO_2_ [21,43,44].

The EGCs were detached from the tissue culture flask with 0.05% trypsin-EDTA and seeded on six-well culture plates at a concentration of 0.5 × 10^6^ cells/well in 2 mL of complete growth medium and allowed to adhere overnight. Then, the EGCs were treated or not with TcdB at 0.1 ng/mL and 10 ng/mL for 24 h at 37 °C with 5% CO_2_.

For all the experiments and analyses, at 24 h, controls and TcdB-treated EGCs were detached as described above and washed. Then cell viability and total cell numbers were determined by erythrosine B dye-exclusion assay.

### 2.3. Flow Cytometry Analysis of Apoptosis and Cell-Cycle Phases

The controls and the TcdB-treated EGCs were recovered and analyzed by flow cytometry to evaluate the DNA content for the detection of cell-cycle changes and apoptosis after 24 h infection with TcdB [21,43,44]. Pellets of the controls and the TcdB-treated EGCs were resuspended in 1 mL of hypotonic fluorochrome solution constitute by propidium iodide (PI) 50 μg/mL, 0.1% Triton X-100 and 0.1% sodium citrate. The samples were incubated for 2 h at 4 °C in the dark, then the PI fluorescence of individual nuclei was measured utilizing a flow cytometer EPICS XL-MCL (Beckman Coulter, FL, USA) [21,43,44]. Apoptosis was analyzed as described by Nicoletti et al. [61]. The data were processed by an Intercomp computer and analyzed with EXPO32 software (Beckman Coulter). The phases of cell-cycle were analyzed by measuring DNA-bound PI fluorescence in the orange-red fluorescence channel (FL2) with linear amplification. The percentage of cells in each cell-cycle phase was examined using ModFit 1.0 software (Verity Software House, Topsham, ME, USA) [21,43,44].

### 2.4. Sample Preparation for Lipidomic Analysis

The controls and the TcdB-treated EGCs were recovered after 24 h infection with TcdB, and lipids were extracted from the pellets using the MMC method (Methanol, MTBE, and CHCl_3_) according to the protocol of Pellegrino et al., with minor modifications [62]. First, 36 mL of the MMC extraction mixture was prepared by mixing 11 mL of methanol, 12 mL of MTBE, 12 mL of CHCl_3_, and 1 mL of Lipidomix Internal Standad (Avanti Polar Lipids, Alabaster, AL, USA) previously diluted 1:10 in methanol. A freshly prepared MMC extraction mixture (1 mL) was added to each pellet. An Eppendorf tube with only the MMC mixture was prepared as a blank. After the addition of MMC, the samples were vortexed for 5 s, immerged for 10 min in an ultrasonic bath, vortexed for 5 s again, and then shaken for 20 min at 1500 at 20 °C in a Thermoshaker (Euroclone, Milan, Italy). The samples were subsequently centrifugated at 16,000 rcf for 10 min at 4 °C, and the resulting supernatant was transferred into a 1.5 mL autosampler vial. A further sample pool was formed by mixing 20 μL of each supernatant in an autosampler vial. All the supernatants were dried at 60 °C with a gentle stream of N_2_. Each residue was resuspended in 50 μL of MeOH/Toluene 9:1 mixture, transferred in an autosampler vial with glass insert, and injected in LC/MS.

### 2.5. LC/MS Analysis

LC/MS analysis was performed with an Agilent analyzer, consisting of an Agilent 1260 Infinity II liquid chromatograph coupled with Agilent 6530 Accurate-Mass Q-TOF (quadrupole-time of flight) analyzer and an Agilent JetStream source. Liquid chromatographic separation was performed by an Ascentis express C18 (150 mm × 2.1 mm, 2.7 µm) Supelco column maintained at 50 °C with a flow rate of 0.25 mL/min. All the LC/MS conditions and row data processing were conducted as previously described [63].

### 2.6. Statistical Analysis

All the data are expressed as mean ± S.D. (standard deviation) of the seven experiments. GraphPad Prism 9.0.0 software was used to make the statistical charts. The Shapiro–Wilk normality test was applied to analyze the data normality distribution. Comparisons of multiple groups were analyzed using one-way ANOVA or two-way ANOVA followed by Tukey’s post hoc test for normally distributed data for evaluating the significance of the differences between the groups. *p*-values = or less than 0.05 were defined as statistically significant.

The statistical analysis of the lipid data matrix was carried out using MetaboAnalyst (5.0) web platform [58]. The lipid pathway analysis and Chain Classification analysis were performed by LipidOne [59].

## 3. Results

### 3.1. TcdB Induces Apoptosis, Necrosis and Cell-Cycle Arrest in EGCs

We previously demonstrated that TcdB induces in EGCs cytopathic and cytotoxic effects in a dose- and time-dependent manner [21]. In fact, we found the following effects of TcdB in EGCs: early cell rounding with Rac1 glucosylation, early G1 and G2/M cell-cycle arrest, and apoptosis by a caspase-dependent but mitochondria-independent pathway [21,43].

In the present study, we analyzed the effects of the 0.1 ng/mL and 10 ng /mL TcdB concentrations on EGC lipidome, after 24 h of infection. We assessed: (a) the percentage of apoptotic cells (hypodiploid DNA content) by flow cytometry, (b) cell viability and the number of live cells by erythrosine B staining of cells, (c) cell-cycle distribution by flow cytometry, and (d) cell lipidomic profiles.

The results obtained confirmed that TcdB induces apoptosis in EGCs in a dose-dependent manner. In fact, we found about 11% apoptosis with 0.1 ng/mL TcdB and 23% apoptosis with 10 ng/mL TcdB (Figure 1A,B). Regarding the percentage of erythrosine B+ cells, we found approximatively a 14.0% of erythrosine B+ cells at 0.1 ng/mL TcdB and approximatively 21% at 10 ng/mL TcdB (Figure 1C), whereas the number of live EGCs treated with TcdB at 0.1 ng/mL and 10 ng/mL was reduced, respectively, by approximately 49%, and 67%, compared to the control EGCs (CTRL) (Figure 1D). In addition, cell-cycle analysis by flow cytometry showed a decrease in S phase cells with both TcdB at 0.1 ng/mL and 10 ng/mL and an accumulation of cells in the 4N peak of flow cytometry, representing both cells in the G2/M phase and bi-nucleated cells in the G1 phase (Figure 1E,F).

These results confirmed that TcdB induces apoptosis and also cell death by necrosis (demonstrated by the reduction in both cell viability and number of live cells). However, the reduced number of live cells observed between the controls and the TcdB-treated EGCs (both doses) is mainly due to the growth inhibition of the EGCs induced by TcdB that was associated with the induction of cell-cycle arrest in the G1 and G2/M phases.

### 3.2. Effect of TcdB Exposure on Lipid Composition of EGCs

Lipid analysis of TcdB-treated and control EGCs was carried out by LC/MS. Figure 2 shows the content of the total detected lipids (panel A) and lipid class distribution (panel B) in the TcdB-treated and the control EGCs. 

The quantitative analysis of the total detected lipids revealed that EGCs treated with the highest concentration of TcdB (10 ng/mL) displayed a significantly higher lipid/protein ratio compared to the controls (Figure 2A). Regarding lipid distribution, the most abundant lipid classes were phosphatidylcholine (PC) (~35% of total lipids), followed by phosphatidylethanolamine (PE) (~25% of total lipids), phosphatidylinositol (PI) (~10% of total lipids), phosphatidylserine (PS) (~10% of total lipids), and sphingomyelin (SM) (~10% of total lipids) (Figure 2B). The comparison of TcdB-treated EGCs and controls revealed differences in lipid class composition. Specifically, the content per μg of protein of all lipid classes, except TG, was higher in EGCs treated with the highest concentration of TcdB as compared to the controls (Figure 2B). A higher level of lyso-PL (i.e., LPC and LPI) was also observed in EGCs treated with 0.1 ng/mL TcdB (Figure 2B).

Then, we reported further comparative analysis of the molecular species within the most abundant lipid classes between the controls and the TcdB-treated EGCs. This is a relevant aspect, considering that membrane properties and lipid-associated signalling pathways are not only affected by the quantity of specific lipid classes but are also influenced by fatty acids chain length and saturation within complex lipids. 

Regarding PE, we detected 3 molecular species of PE-O (alkyl- phosphatidyethanolamine, PE with a fatty acid at the sn-1 position linked by a ether linkage and the fatty acid at the sn-2 position linked by an ester linkage to the glycerol moiety) (representing ~0.4% of total PE), 26 molecular species of PE-P (phosphatidylethanolamine plasmalogens, PE with a fatty acid at the sn-1 position linked by a vinyl ether linkage and the fatty acid at the sn-2 position linked by an ester linkage to the glycerol moiety) (representing ~40% of total PE), and 13 species of PE (diacyl-phosphatidylethanolamine) (Figure 2). Statistical analysis revealed that levels of most molecular species of PE-diacyl were increased in EGCs treated with 10 ng/mL TcdB, compared with controls (Figure 3). Interestingly, these differences accounted for the greater content of the total PE observed in EGCs treated with 10 ng/mL TcdB (Figure 2). Moreover, the level of 10 out of 26 molecular species of plasmenyl-PE were also present at a higher level in EGCs treated with 0.1 ng/mL TcdB, compared with the controls (Figure 3). In EGCs treated with 10 ng/mL TcdB, we also observed increased levels of the unique annotated LPE (18:1) (Figure 2).

The analysis of the lipids containing choline as polar head annotated 17 PC, 1 PC-P, 1 oxydated PC (PC 16:0,O1_18:1:O1), and 1 LPC (16:0) in the EGCs. The exposure of the EGCs to 10 ng/mL TcdB significantly increased the levels of 13 molecular species of PC (Figure 4A), accounting for the higher content of total PC observed in the treated EGCs compared with the controls (Figure 2B). The content of LPC 16:0 was increased by both concentrations of TcdB (Figure 2B). 

In the EGCs, we annotated 6 PSs, and 3 of them were present at higher level in EGCs treated with 10 ng/mL of TcdB with respect to the controls (Figure 4B).

Regarding PI, 15 of the 18 annotated species were increased in the EGCs treated with 10 ng/mL TcdB (Figure 5A). The exposure of EGCs with 10 ng/mL TcdB also increased the level of the unique PI-O (Figure 5A) and of all the LPI species (Figure 5B). The levels of 7 PI and of all LPI species were also increased in the EGCs treated with 0.1 ng/mL TcdB (Figure 5A,B).

In the EGCs we detected 6 molecular species of SM (~10% of total detected lipids), and 2 of them (SM 16:1; 2O/26:1; SM 18:1; 2O/24:1) were increased upon exposure to the highest concentration of TcdB.

To evaluate the impact of acyl chain changes on the physico-chemical properties of membranes, we analyzed the length and saturation degree of acyl chains. TcdB at 10 ng/mL significantly increased the level of lipids containing saturated (SFA), monounsaturated (MUFA), and polyunsaturated (PUFA) fatty acids (Figure 6A). Compared to the controls, the EGCs treated with 10 ng/mL TcdB displayed higher levels of PL species containing long fatty (LCFA) and very long fatty acid chain (VLCFA) (Figure 6B).

PCA and heatmap were applied to analyse the three experimental groups based on the whole lipid data (Figure 7). PCA analysis showed that the three lipidomic datasets clustered into two main groups. EGCs treated with the two concentrations of TcdB were not clearly separated from each other but were clearly separated from the controls, which clustered in a separate group (Figure 7A). The heatmap reported in Figure 7B was built using the 25 lipids that were most significant in the ANOVA test. The hierarchical analysis showed that the EGCs treated with TcdB (both 0.1 and 10 ng/mL) clustered separately from the controls. The 14 lipid species underexpressed in the TcdB-treated EGCs included 3 PSs, 3 PEs, and 4 SMs. The remaining 10 lipids overexpressed in the TcdB-treated EGCs were predominantly LPI and PI (Figure 7B).

The lipid class composition comparison between the 10 ng/mL TcdB-treated EGCs and the controls was carried out with LipidOne (Figure 8A), a bioinformatic tool tailored to analyze possible pathways underlying detected lipidomic changes. The results allowed us to hypothesize that the changes in lysoPL levels observed in the TcdB-treated EGCs might be caused by an increased expression/activation of phospholipase A (PLA) having as substrates PI, PE, and PC. The increased levels of plasmenyl-PE in the TcdB-treated EGCs could possibly be due to plasmenylethanolamine desaturase 1 (PEDS1), an enzyme converting alkyl-PE to plasmenyl-PE (PE-O to PE-P). This analysis also indicates a possible reduction in the expression/activity of phospholipase C (PLC) having PI as a substrate, and of lysophosphatidylcholine acyltransferase 1 (LPCAT), an enzyme that catalyzes the conversion of LPC to PC in the presence of acylCoA.

The analysis performed with LipidOne on individual molecular species provided information regarding the de novo synthesis/interconversion reactions that might be responsible for the acyl chain remodeling of lipids in EGCs treated with 10 ng/mL TcdB (Figure 8B). In particular, the analysis indicated an increased expression/activity of: (i) PDES1 acting on PE-O 18:1_22:4; (ii) phosphatidylethanolamine N-methyltransferase (PEMT), converting PE 18:0_18:1 into the corresponding PC by methylation; (iii) sphingomyelinase (SMase) hydrolyzing SM 18:1;2O/24:0 to produce Cer and phosphorylcholine; (iv) PLC acting on PS 18:0_18:1 and PE 18:0_18:1 to generate the corresponding DGs (Figure 8B). On the opposite side, other lipid changes indicated the reduced expression/activity of: (i) ethanolamine phosphotransferase 1 (EPT1), catalyzing the transfer of phosphoethanolamine from CDP-ethanolamine to DG 16:0_18:1 and DG 10:0_18:1, thus producing the corresponding PEs; (ii) PS synthase 1 (or serine exchange enzyme) (PSS1), catalyzing the formation of PS from PC or PE (PC 18:0_18:1 into PS 18:0_18:1); and (iii) PLC acting on PI 16:0_18:1, generating the corresponding DG. As for the acyl chain composition, the analysis performed with LipidOne indicated an increased expression/activity of desaturase 5 (D5D) and elongases 1,2,3,5 (ELOVL1,2,3,5) (Figure 8C).

## 4. Discussion

The present study demonstrated that TcdB induces changes in the phospholipidome of EGCs, which might represent one of the events underlying the cytopathic and apoptotic effects induced by TcdB in EGCs [21,43]. Although signaling pathways underlying harmful effects induced by TcdB have been extensively described in EGCs [21,43] and other cell types [15,16,17,22,23,24], to date no studies have investigated lipid changes induced by TcdB or TcdA in EGCs or other cell types. Here, we demonstrated that the exposure to the highest concentration of TcdB (10 ng/mL) induced in EGCs relevant modification of phospholipid profiles. Changes in phospholipid class levels were so relevant as to allow the clusterization of TcdB-treated and control cells into separate groups. The major determinants for the different lipidome of TcdB-treated vs. control cells were PI, PE-P and lyso-PL. Moreover, TcdB also induced rearrangements of acyl chains within phospholipid classes. 

Several bacteria and their toxins, independently from the modes of interaction with target cells, modify the lipid composition of host cell plasma and intracellular membranes by acting on the expression/activity of lipid metabolism enzymes. This evidence suggests that pathogens or their toxins induce modifications of lipid asset and trigger signaling processes in host cells to allow their internalization, survival, and replication [52,55].

The first interesting result reported is that treatment with the highest concentration of TcdB (10 ng/mL) induces an increase in all detected lipid classes except TGs, whereas increased levels of LPI and LPC were also present in EGCs treated with 0.1 ng/mL TcdB. Consistently, an increased level of several classes and subclasses of lipids has been observed during gut colonization and infection in a mouse model of CDI [64]. 

Notably, PCA performed using global lipid datasets revealed that differences between TcdB-treated and control EGCs were significant enough to cluster the samples into two distinct groups (Figure 6A). Hierarchical analysis showed that among 25 lipid molecular species responsible for the clustering, 10 are phospholipid species containing inositol as polar head (PI and LPI), which were all overexpressed in TcdB-treated EGCs (Figure 6B). 

Interestingly, phospholipid classes responsible for clusterization are not only relevant constituents of membrane architecture but also play pivotal roles in several biological processes. PIs and their polyphosphorylated forms are involved in signal transduction, cytoskeleton reorganization, membrane dynamics, and membrane and vesicular trafficking [65]. In particular, the polyphosphorylated PIs mediate signal transduction pathways as a source of second messengers and modulate signaling proteins activity and/or recruitment to the membrane [65]. The level of PIs and their polyphosphorylated forms in membrane microdomains regulate cell death processes through cytoskeleton rearrangements and membrane recruitment/activation of death effectors [65]. The concentration of polyphosphorylated PIs in membranes is regulated by the activity of kinases that phosphorylate the inositol ring (i.e., PI-4 kinase and PI-4,5Pkinase) and phosphatases that remove the phosphate group. In the context of the present study, it is interesting to note that TcdB induces inactivation of the Rho family GTPases [15,16,17,21], which causes an inhibition of PI4,5-kinase, thus reducing PI(4,5)P2 levels [66]. Changes in the levels of PI observed in EGCs treated with TcdB indicate that this phospholipid class is a target of TcdB which is able not only to modify the levels of PI phosphorylated forms [67,68,69] but also to increase the total content of this phospholipid class. 

Plasmalogens are a subclass of glycerophospholipids characterized by a vinyl ether (alkenyl) and an ester bond at the sn-1 and sn-2 position of glycerol, respectively. They are enriched in PUFA and their concentration in membranes depends on the tissue [70]. Plasmalogens have many effects on the physical properties of biological membranes, as they modulate membrane fluidity and stabilize the formation of membrane domains or negatively curved surfaces. In addition, their role in signal transduction is also reported. For instance, the plasmalogen-linked PUFA (i.e., arachidonic and docosahexaenoic acids) are precursors of bioactive lipids (i.e., eicosanoids and docosanoids), involved in inflammation [70]. Another important aspect is that plasmalogens are endogenous antioxidants, acting as scavengers of reactive oxygen species and reactive nitrogen species [71,72,73]. In fact, the vinyl ether linkage scavenges oxidative radicals to protect membrane lipids from peroxidation. This property of plasmalogens could explain the anti-inflammatory and antiapoptotic effects of exogenous PE-P in human intestinal tract cells [74]. The increased levels of PE-P observed in EGCs treated with TcdB might represent a mechanism employed by EGCs to mitigate the cytotoxic effects induced by TcdB.

Lyso-PL levels are also increased in TcdB-treated EGCs. Lyso-PLs participate in cell signaling, as they are considered bioactive lipids as well as precursors of other lipid mediators including PAF and endocannabinoids [75]. They are produced by PLA by membrane glycerophospholipids hydrolysis, a reaction also producing free fatty acids. Then, they can be also converted again into glycerophospholipids by LPCATs in the presence of acyl-CoA. The coordinated action of PLAs and LPCATs generates glycerophospholipids by a pathway called “Land’s cycle”, responsible for the rapid remodeling of lipid membrane occurring during important cellular events, such as the activation of signal transduction pathways, cell proliferation and death. Therefore, the proper regulation of the Land’s cycle is important to control the accumulation of potentially toxic lipid and to maintain the integrity of membranes [76].

The bioinformatic analysis of lipid datasets indicates lipid metabolism enzymes possibly involved in TcdB-treated EGCs changes. The most relevant predictions consisted in the possible increased expression/activation of PLA, PEDS1, D5D, and ELOVL2,3,5, and the reduced expression/activation of EPT1 and PSS1. The higher content of lyso-PLs and phospholipids containing UFA, LCFA, and VLCFA, together with the predicted increased expression/activity of PLA, indicated that in TcdB-treated EGCs these species might not derive from the de novo synthesis but from phospholipid remodelling pathways. This hypothesis was supported by the predicted reduced expression/activity of two enzymes involved in the de novo synthesis of phospholipids, such as EPT1 and PSS1.

Few studies have previously demonstrated that TcdB modulates in host cells the activity of phospholipid metabolizing enzymes, such as PLD, PLC, PLA_2_ and PI4,5 kinase [77,78,79,80,81,82]. A study demonstrates that in fibroblasts TcdB induces activation of PLC and PLA_2_ [77,78,79]. In HL60 and HEK cells, the TcdB-induced cytotoxicity and cytoskeletal rearrangements are associated with the inhibition of PLD [80,81,82], caused by the Rho-dependent depletion of membrane PIP_2_, a cofactor of PLD [81].

In conclusion, our results demonstrate that TcdB induced relevant changes in the phospholipidome of EGCs. Interestingly, the lipid classes whose levels were modified are important determinants of membrane properties (i.e., fluidity, lipid rafts formation) and play significant roles in cell signaling. While the lipidomic analysis allowed us to identify which lipid classes were modified by TcdB, the bioinformatic analysis carried out subsequently allowed us to predict the lipid metabolism pathways that might be altered by TcdB. Currently, we cannot establish if changes in lipid profile represent one mechanism used by the TcdB to modulate host cell signaling pathways and consequently favour *C. difficile* pathogenicity, or if these changes represent a mechanism used by host cells to survive and mitigate the cytotoxic effects exerted by TcdB. However, the results of this study suggest that *C. difficile* might be able to alter lipid metabolic homeostasis in host cells. 

The detailed view of lipidome of TcdB-treated EGCs might be considered a fingerprint representative of lipid profiling of gut cells during CDI. Since lipidome is an important indicator for the metabolic/pathological state of a cell, the elucidation of changes induced by *C. difficile* toxins in the host lipid asset might represent an important step in the drug development process.

## Figures and Tables

**Figure 1 cells-13-01103-f001:**
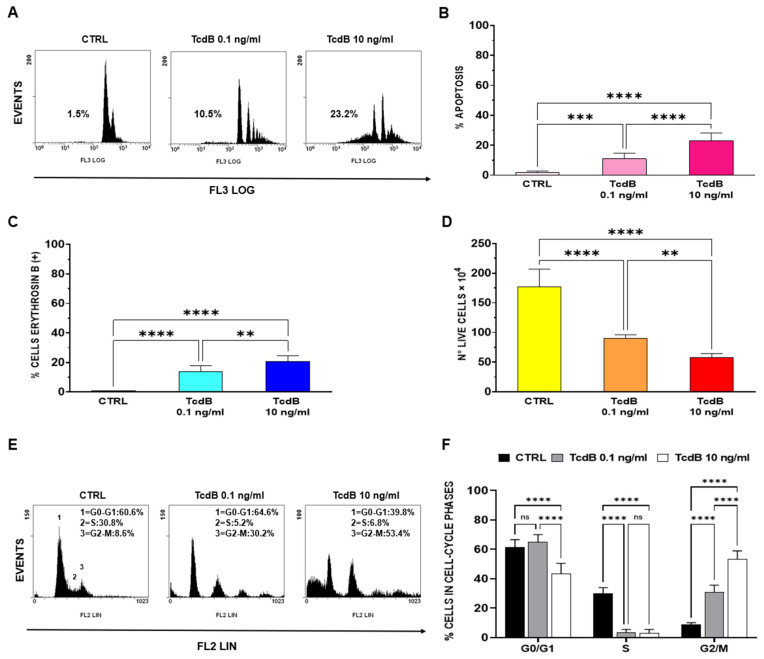
Effect of TcdB on EGC apoptosis, cell viability, cell growth, and cell-cycle phases. Control EGCs (CTRL) or EGCs treated with TcdB at 0.1 ng/mL or at 10 ng/mL were recovered at 24 h, and the following was determined: (**A**,**B**) Apoptosis measuring the percentage of hypodiploid nuclei by flow cytometry; the percentage of erythrosine B (+) cells (**C**) and the number of live cell (**D**) by erythrosine B (+) dye-exclusion assay; the percentages of cells in the cell-cycle phases by flow cytometry with ModFit software (**E**,**F**). In (**A**) are shown the DNA fluorescence flow cytometric profiles with percentages of hypodiploid nuclei of one experiment, representative of seven. In (**E**) are shown the flow cytometric profiles with cell percentages in G0/G1, S and G2/M of one experiment representative of seven. In (**B**–**D**,**F**) are shown the graphs, and the data are the mean ± S.D. of seven experiments. (**B**–**D**), Statistical analysis was performed by one-way ANOVA and Tukey’s multiple comparisons test, (**F**), Statistical analysis was performed by two-way ANOVA and Tukey’s multiple comparisons test. ** *p* < 0.01, *** *p* < 0.001, **** *p* < 0.0001, ns *p* > 0.05. Abbreviations: *Clostridioides difficile* toxin B (TcdB), enteric glial cells (EGCs), control EGCs (CTRL), standard deviation (S.D.).

**Figure 2 cells-13-01103-f002:**
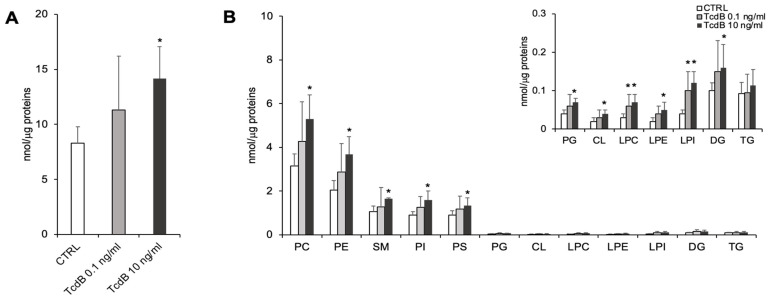
Lipid composition of TcdB-treated and control EGCs. Lipid extracts from CTRL and TcdB-treated EGCs have been analyzed by LC-MS/MS. (**A**) Quantity of total lipids (sum of all annotated species) relative to protein content. (**B**) Amount of each lipid class (sum of all annotated species belonging to a specific class) is expressed as nmol/μg of proteins. Data are reported as mean ± S.D. Statistically significant differences resulted by *t*-test (* *p* < 0.05, ** *p* < 0.01, TcdB-treated EGCs vs. CTRL) (n = 7).

**Figure 3 cells-13-01103-f003:**
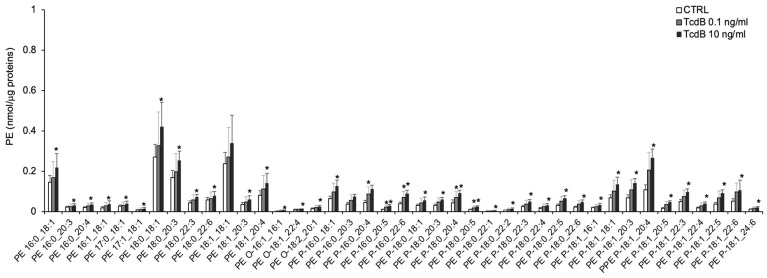
Molecular species of PE, PE-O, and PE-P detected in TcdB-treated and control EGCs. Lipid extracts from CTRL and TcdB-treated EGCs were analysed by LC/MS-MS. Data, expressed as nmol of lipid species/μg of proteins, are reported as mean ± S.D. Statistically significant differences resulted by *t*-test (* *p* < 0.05, TcdB-treated EGCs vs. CTRL) (n = 7).

**Figure 4 cells-13-01103-f004:**
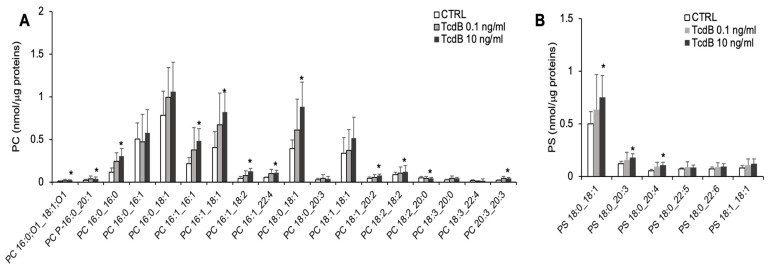
Molecular species of PC and PS detected in TcdB-treated and control EGCs. Lipid extracts from CTRL and TcdB-treated EGCs were analysed by LC-MS/MS. (**A**) Molecular species of PC. (**B**) Molecular species of PS. Data, expressed as nmol of lipid species/μg of proteins, are reported as mean ± S.D. Statistically significant differences resulted by *t*-test (* *p* < 0.05, TcdB-treated EGCs vs. CTRL) (n = 7).

**Figure 5 cells-13-01103-f005:**
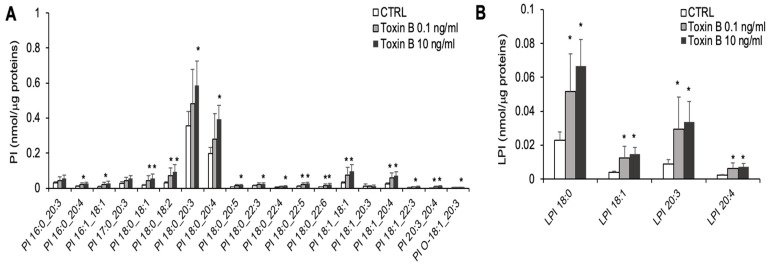
Molecular species of PI and LPI detected in TcdB-treated and control EGCs. Lipid extracts from CTRL and TcdB-treated EGCs were analyzed by LC/MS-MS. (**A**) Molecular species of PI. (**B**) Molecular species of LPI. Data, expressed as nmol of lipid species/μg of proteins, are reported as mean ± S.D. Statistically significant differences resulted by *t*-test (* *p* < 0.05, TcdB-treated EGCs vs CTRL) (n = 7).

**Figure 6 cells-13-01103-f006:**
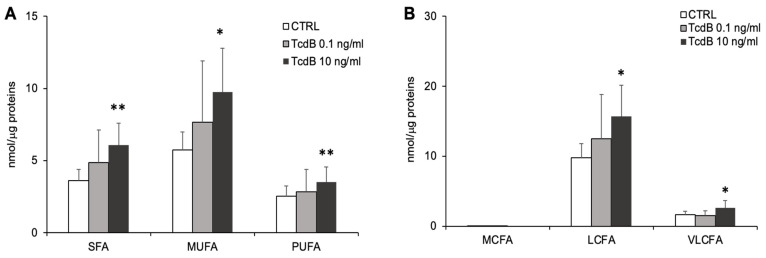
TcdB-induced changes in saturation and length of acyl chains forming complex lipids in TcdB-treated and control EGCs. (**A**) The graph reports the sum of all saturated (SFA), monounsaturated (MUFA), and polyunsaturated fatty acids (PUFA) forming complex lipids. (**B**) The graph reports the sum of all medium (MCFA, <12 carbons), long (LCFA,12–20 carbons), and very long acyl chain (VLCFA, 20–36 carbons) forming complex lipids. Data, expressed as nmol of lipid species/μg of proteins, are reported as mean ± S.D. Statistically significant differences resulted by ANOVA (* *p* < 0.05 TcdB-treated EGCs vs. CTRL; ** *p* < 0.01 TcdB-treated EGCs vs. CTRL) (n = 7).

**Figure 7 cells-13-01103-f007:**
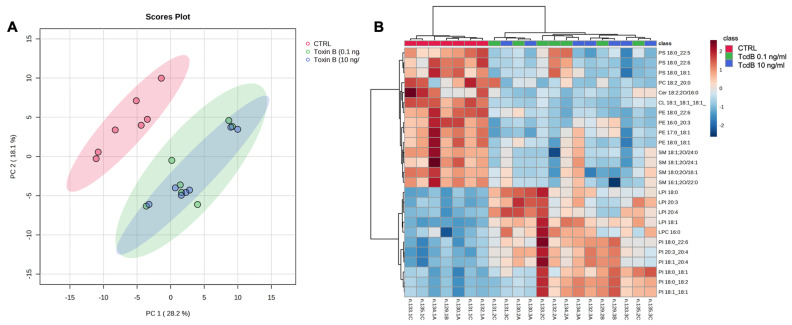
Multivariate statistical analyses of the lipidomic datasets. (**A**) PCA analysis of all lipid species annotated in CTRL and TcdB-treated EGCs after median normalization and Auto scaling. CI 95% ellipses are shown for the three different groups. (**B**) Heatmap of 25 most significant (ANOVA) lipids. Lipids and samples were both ordered using hierarchical clustering (Pearson distance). Autoscaled relative abundances of lipid species are represented by graduation of color from red (most abundant) to blue (least abundant). These statistical analyses were performed using Metaboanalist 5.

**Figure 8 cells-13-01103-f008:**
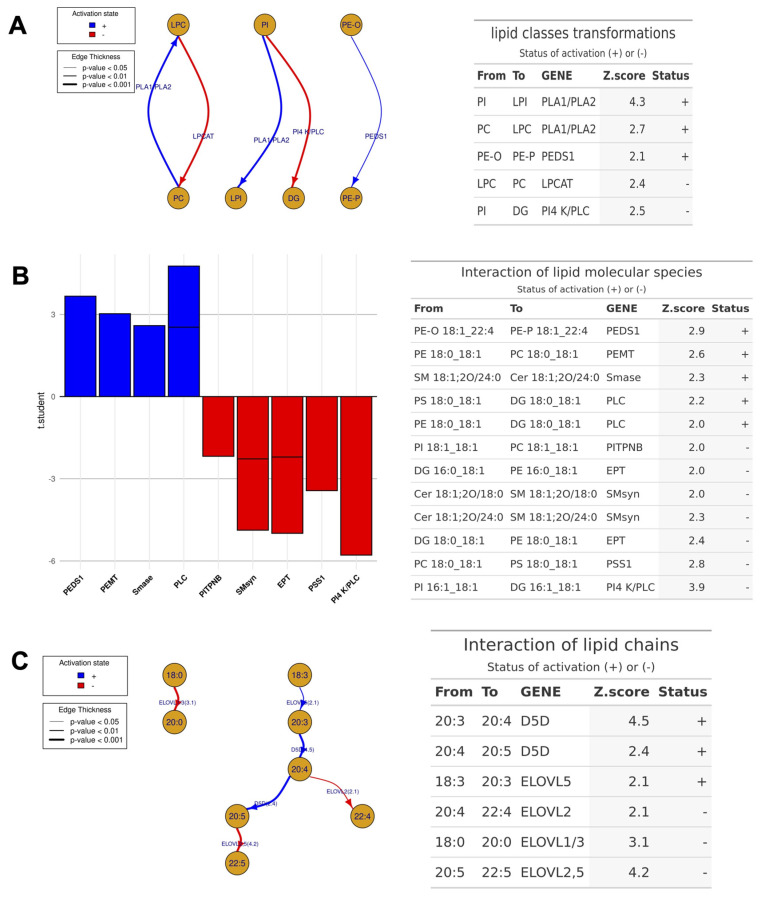
Lipid Pathway Analysis. The potential reactions among lipids in the TcdB 10 ng/mL-treated EGCs compared to the CTRL were investigated using LipidOne at the class level (**A**), lipid molecular species level (**B**), and lipid building blocks level (Acyl, Alkyl, or Alkenyl Chains) (**C**). Schematics (**A**,**C**) depict network diagrams, while B displays a bar graph representing the predicted activation status of genes/enzymes. Blue and red represent the +/− activation status, respectively. Each schematic is accompanied by a table detailing individual reactions, the involved genes, Z-score, and activation status.

## Data Availability

Data are contained within the article.

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
