# Peer review of "PhosphoLipidome Alteration Induced by Clostridioides difficile Toxin B in Enteric Glial Cells"

_cells, 2024, doi:10.3390/cells13131103_

Round 1

Reviewer 1 Report

Comments and Suggestions for Authors

In this manuscript (ID# cells-3057988), entitled “Lipidome Alteration Induced by Clostridioides difficile Toxin B in Enteric Glial Cells”, authors Buratta et al have studied the effect of Clostridioides difficile Toxin B (TcdB) on lipid profile in enteric glial cells (EGC) using LC-MS/MS. Their results demonstrate that TcdB treatment alters the lipidome of EGC. The results from this study are relative novel. However, there are several major concerns which are listed in the following paragraphs:

1) Results from this study indicate that treatment with TcdB alters the lipid profile in EGCs. However, the role of those lipid alterations in the TcdB-induced apoptosis and cytotoxicity is unclear. It is uncertain that those lipid alterations contribute to the TcdB-induced cytotoxicity in EGCs. The missing experiment significantly lowers the significance of this study.

2) The representative images of flow cytometry should be provided in the Figure 1.

3) In Figure 2, please provide statistical analysis. If no significant, please state so.

4) Quality of figures should be improved in order to be seen clearly. The font in some figure is too small to be read. The font should be kept consistently among the figures.

5) Please provide information regarding the labels of X-axis in Figure 4. For example, what is “PC 16:0,O1_18:1:O1?       

6) The writing language should be edited thoroughly including grammar and spellings. Please define “ECG” “ECGc”; Are they EGCs?

Comments on the Quality of English Language

1) Check gramma and spelling thoroughly.

2) Abbreviations should be spelled out at the first time when they present in the text. 

Reviewer 2 Report

Comments and Suggestions for Authors

The work by Buratta and colleagues entitled “Lipidome Alteration Induced by Clostridioides difficile Toxin B 2 in Enteric Glial Cells” describes the signature of 12 lipid classes in glial cells and the modifications to the lipid composition after 24h exposure to Clostridioides difficile toxin TcdB using 2 different concentrations (0.1 ng/mL and 10 ng/mL). Based on the alterations to the lipid composition and pathway analysis carried out on the lipidomic dataset, the authors conclude that altered levels to specific lipid species provide a fingerprint representative of altered metabolism in TcdB-treated cells and might be used in drug development process. Given the serious threat of antimicrobial resistance (AMR) to global health, an in-depth knowledge on the molecular mechanisms governing the many stages in pathogen infection (adhesion, invasion, and proliferation) in host cells is pivotal, pointing to the validity of the topic here covered.

However, certain aspects in the text and experimental design are ambiguous or lack clarity (please see below) and for these reasons the work cannot be published in its present form.

Major points:

1.      The authors have profiled 12 lipid classes (PC, PE, PS, PI, PG, LPC, LPE, LPI, CL, SM, DG and TG). While the classes screened include phospholipids (PL), glycerolipids (GL) and sphingolipids (SL) enough to claim that they looked at the lipidome (as stated in the title), the reality is that the lipidome is much broader than that here covered and important classes such as free fatty acids (FFA), sterols (Chol and CE) and glycosphingolipids (sulfatides, HexCer, …) were not included. Moreover, the bulk of the work (and Discussion Section, pg 11-13) is mostly focused on the alterations to the phospholipidome. In view of this, the title should reflect the work and findings and consider changing to “(Phospho)Lipidome Alteration Induced by Clostridioides difficile Toxin B 2 in Enteric Glial Cells”.

2.      As stated by the authors, changes to the lipid composition impact the physical properties of biological membranes. However, one of the major modulators of membrane fluidity in eukaryotic cells is cholesterol (please see work by Maxfield and van Meer, Curr Opin Cell Biol (2010) 22, 422; Norma et al., Cell Mol Bioeng (2010) 3, 151), which remarkably was not included in the lipid classes analysed. This is an additional aspect to reinforce the change of the title.

3.       Figure 6 (page 8) describes the changes to saturation and chain length in the most abundant glycerophospholipids (and forming the bulk of cell membranes) in TcdB-treated glial cells compared to non-treated. While it is obvious that the first point of contact of pathogens to cells is the cell (plasma) membrane and thus adaptation of the cell to the stress stimuli with modification of membrane lipids is likely to occur at the plasma membrane, it is not clear why the authors have translated the whole glial cells lipidome to the plasma membrane considering that no organelle sub-fractionation protocols were carried out or even isolation of the glial cell membrane (see Experimental Section). To avoid misleading conclusions or misinterpretation, this aspect needs to be clarified throughout the text.

4.      The lipidomic dataset was acquired using the LC-MS conditions as described in Alabed et al., Biomol (2023) 13,1. However, as the authors have only focused on the phospholipidome it is not clear why the authors acquired full MS in the range m/z 50-3200, as FA were not included in their list of lipids and lyso-lipids have a typical m/z value starting from 400-580 a.m.u. and cardiolipins have a typical m/z value up to 2000 a.m.u.

Other points:

1.      In their list of cited references, the authors have included 11 references of their work (13%). While the authors may be a leading reference in the field of Clostridioides difficile infection and have teamed up with others with lipidomics expertise (Roberto Pelegrino), this high number of own references may be viewed as a way to increase citations of their work (and indirectly raise their own profile).

Round 2

Reviewer 1 Report

Comments and Suggestions for Authors

no further recommendation.